# Efficient Micropropagation of *Sedum sediforme* and *S. album* for Large-Scale Propagation and Integration into Green Roof Systems

**DOI:** 10.3390/plants14121819

**Published:** 2025-06-13

**Authors:** Ignacio Moreno-García, Begoña García-Sogo, Salvador Soler, Adrián Rodríguez-Burruezo, Vicente Moreno, Benito Pineda

**Affiliations:** 1Instituto de Biología Molecular y Celular de Plantas (IBMCP), Universitat Politècnica de València (UPV), Camino de Vera s/n, 46022 Valencia, Spain; nachomoreno21@gmail.com (I.M.-G.); btgsogo@gmail.com (B.G.-S.); vmoreno@ibmcp.upv.es (V.M.); 2Instituto Universitario de Conservación y Mejora de la Agrodiversidad Valenciana (COMAV), Universitat Politècnica de València (UPV), Camino de Vera s/n, 46022 Valencia, Spain; salsoal@btc.upv.es (S.S.); adrodbur@upvnet.upv.es (A.R.-B.)

**Keywords:** micropropagation, axillary shoot culture, somatic organogenesis, Crassulaceae

## Abstract

Urban expansion has led to two significant environmental challenges: the reduction in green spaces and the rise in urban temperatures, decreasing city livability. Green roofs have emerged as a sustainable solution to mitigate these issues, offering ecological and economic benefits while improving building energy efficiency. Some species of the genus *Sedum*, particularly *Sedum sediforme* and *Sedum album*, are ideal for such green infrastructure due to their non-aggressive and superficial root system, high drought tolerance, low nutrient needs, pest and disease resistance, and metabolic adaptability during dry periods. This study aims to optimize the large-scale production of two native ecotypes of *S. sediforme* and *S. album* from the Valencian Community through an efficient propagation system that enables uniform plant production in limited space. For this purpose, we have developed micropropagation systems that allow a rapid multiplication of these two species. A direct morphogenesis system was established using axenic plant shoots, and a protocol for adventitious organogenesis from leaves was also developed. These methods significantly enhance propagation speed, spatial efficiency, and plant uniformity. Notably, the metabolic plasticity of *S. sediforme* and *S. album* reduces abiotic stress during acclimatization, promoting efficient ex vitro establishment and functional integration into extensive green roof ecosystems.

## 1. Introduction

Green roofs have emerged as a vital element in sustainable urban development due to their numerous ecological and economic benefits. Broadly, they are categorized into two types based on the thickness of the substrate layer: intensive and extensive systems. Intensive green roofs feature a soil depth greater than 15 cm, allowing for cultivating a wide variety of plant species. The required soil depth, irrigation needs, and maintenance levels depend on the selected vegetation; however, these systems generally demand significant maintenance [1]. In contrast, extensive—or ecological—green roofs are characterized by a shallow substrate and low-maintenance vegetation. These systems are considerably lighter and typically do not require irrigation unless prolonged drought conditions occur [1].

In this global context of climate crisis, extensive green roofs are much more interesting than intensive ones since the vegetation used requires little maintenance and can withstand long periods of drought. Long-term studies on the maintenance of green roofs indicate that it is essential to identify planting traits that will endure, thus reducing replanting and other environmental costs [2]. Extensive green roofs typically incorporate a relatively limited range of plant species, with *Sedum* species being the most commonly utilized due to their resilience and low maintenance requirements [3,4]. Most *Sedum* species possess a set of characteristics that make them particularly advantageous for use in extensive green roofs or ground covers: they utilize the Crassulacean Acid Metabolism (CAM) photosynthetic pathway, have shallow root systems, exhibit high drought tolerance, require minimal nutritional requirements, and have a high reproductive capacity [4]. In this sense, *Sedum* species buffer against major changes to environmental conditions or abrupt changes to maintenance, adding insurance against the failure of extensive green roofs [5].

Micropropagation is one of the most significant techniques in plant tissue culture due to its capacity to rapidly multiply selected plant genotypes using minimal starting material. In contrast to conventional propagation through seeds or vegetative methods, micropropagation enables large-scale production of numerous plants, which has led to its widespread application in both research and commercial settings [6]. Additionally, it facilitates the swift generation of genetically identical plants, ensuring uniformity and high quality [7]. Beyond its utility for mass propagation, micropropagation serves as a foundational tool for conserving genetic resources and maintaining biodiversity, particularly in the face of changing climatic conditions [8]. It also offers the advantage of producing a desired number of plants while maintaining them in vitro, free from diseases and harmful pests [9].

Micropropagation is typically carried out via organogenesis or somatic embryogenesis [10]. The optimization of these regeneration techniques has significantly contributed to the development of new or improved varieties through selective breeding and genetic engineering [7]. Ultimately, ongoing advancements in micropropagation and genetic modification are expected to yield increasingly diverse and resilient plant varieties [7]. Efficient micropropagation protocols have been established for various *Sedum* species. For instance, successful protocols have been reported for *Sedum drymarioides*, an endangered and rare species [9]; *Sedum sarmentosum*, commonly used in succulent gardens and as an ornamental plant [11,12]; *Sedum alfredii*, a recently identified Zn/Cd hyperaccumulator [13]; *Sedum spectabile*, a drought-tolerant herbaceous perennial [14]; *Sedum praealtum*, known for its anti-inflammatory and analgesic properties [15]; *Sedum dasyphyllum*, a rare medicinal species [16]; and *Sedum aizoon*, valued for its content of flavonoids and phenolic acids [17].

In this work, we used two species of the genus *Sedum*: *Sedum sediforme* and *Sedum album*. *S. sediforme* is a small, succulent perennial belonging to the Crassulaceae family. Native to the Mediterranean region, it is also found across temperate and cold zones in both hemispheres. This species thrives in environments with minimal soil, often colonizing rooftops, dry stone walls, and rock crevices [18]. It exhibits an upright growth habit and typically reaches a height of 10 cm [19]. Its leaves are fleshy, pointed, oblong, and display a glaucous blue-green hue. *S. album* is a fleshy, perennial herb that grows to approximately 8–20 cm in height [20]. It develops a dense, mat-forming vegetative cover as a result of its persistent shoot regeneration from serpentine, creeping roots [21]. The leaves are sessile, alternate, glabrous, nearly cylindrical, and exhibit a reddish-green coloration [22].

The primary objective of this study was to develop an efficient micropropagation protocol for various ecotypes of *S. sediforme* and *S. album*, both native to the Valencian Community. To the best of our knowledge, no such protocol has previously been reported for either species. The natural propagation of these species is performed by seeds or vegetatively, either as fragments of leaves and plant propagules, or stem fragments [19]. While propagation itself is not particularly challenging, traditional methods are slow and demand a substantial number of parent materials. In contrast, micropropagation enables the large-scale production of uniform, disease-free plants. This approach is especially advantageous for generating clean planting stock, which is critical when introducing species to new environments or supplying commercial nurseries. Furthermore, establishing reliable in vitro regeneration systems for *S. sediforme* and *S. album* paves the way for genetic improvement strategies, including gene transfer and genome editing. In light of these considerations, this study proposes micropropagation protocols as a viable alternative to conventional propagation techniques for both species.

## 2. Results and Discussion

### 2.1. In Vitro Introduction of S. sediforme and S. album Plants

In vitro introduction of the four *Sedum* ecotypes was performed by disinfecting stem segments with commercial bleach, previous pretreatment with running water, and tween 20 detergent, as described in Section 3.1 of Material and Methods. After the surface sterilization process, stem segments were cut with the aid of tweezers and a scalpel into smaller stem explants containing at least one axillary bud. After 15 days, the percentage of success in the sterilization process was evaluated. The best results were obtained in ecotype EC5 of *S. sediforme*, in which an in vitro introduction of about 65% was achieved and in ecotype EC95 of *S. album*, with 61% implantation success (Table 1). In ecotypes EC59 of *S. sediforme* and EC85 of *S. album*, 41% and 46% of implantation success were obtained, respectively (Table 1). It should be noted that the percentage of dead but uncontaminated explants due to the surface sterilization process was nil in the four ecotypes. A total of 82 explants of *S. sediforme* (50 of ecotype EC5 and 32 of ecotype EC59) and 83 of *S. album* (36 of ecotype EC85 and 47 of ecotype EC95) were successfully introduced. From these explants, axenic plants were developed and used for the multiplication experiments by axillary shoot culture and regeneration via somatic organogenesis.

It has been reported that the in vitro establishment of nodal segments from certain species, such as *Cnidoscolus aconitifolius* (Mill.) I.M. Johnst, a multipurpose woody plant, involves a complex and labor-intensive process [23]. In this regard, the authors found that optimal in vitro establishment of this species was achieved following a pretreatment with tap water, soaking in 75% ethanol for 50 s, immersion in 0.1% mercuric chloride for 10 min, and the use of a culture medium supplemented with 3 mL/L of Plant Preservative Mixture (PPM). In contrast, our findings indicate that the in vitro establishment methodology for *S. sediforme* and *S. album* is relatively simple, as it only requires a pretreatment with running water and detergent, followed by surface sterilization with commercial bleach.

### 2.2. Multiplication of S. sediforme and S. album by Axillary Shoot Culture

In order to know the multiplication rate that could be obtained in the ecotypes of both species, the duration of a culture cycle was set at 30 days. The response of the four ecotypes was compared using three different media: BM2, BM1, and BM0 (see Section 3.2 of Material and Methods). We observed that no significant differences were associated with the culture medium for any of the ecotypes. Indeed, each ecotype exhibited similar development in the three culture media (Table 2). Although the differences were not statistically significant, we observed that all the ecotypes seemed to develop slightly better in the medium with lower nutritional inputs (i.e., BM0).

In the ornamental species *Lophophora williamsii* (Lem.) Coult., which exhibits CAM, low sucrose concentrations in the culture medium (15 g/L) have been shown to promote better seedling growth, whereas higher concentrations (30 g/L) have detrimental effects [24]. The authors suggested that excessive sucrose levels might create an unfavorable osmotic gradient, restricting seedling development in this species. Similarly, in some species of the genus *Capsicum*, both immature advanced embryos and mature embryos have demonstrated improved germination efficiency and further growth when cultured in media with moderate sucrose levels [25]. Additionally, studies have shown that culture media supplemented with half-strength MS [26] yield better results than those with full-strength MS. For instance, in the endemic medicinal plant *Ranunculus wallichianus* Wight & Arn, the highest percentage of yellowish-green compact nodular callus formation was observed in a half-strength MS medium compared to a full-strength MS medium [27].

On the other hand, significant differences were observed between ecotypes. Thus, ecotypes EC59 and EC85, among which no significant differences were observed, developed more leaves and reached a higher height than ecotypes EC5 and EC95 (Table 2). Significant differences were also observed between ecotypes EC5 and EC95. In particular, the plant height of ecotype EC5 was significantly lower than that of ecotype EC95, but the number of leaves developed was significantly higher (Table 2).

We performed a new experiment to verify the reproducibility of the results. After a 30-day culture cycle, it was found that six axillary buds could be obtained from ecotypes EC59 of *S. sediforme* and EC85 of *S. album*, while five axillary buds were obtained from ecotypes EC5 of *S. sediforme* and EC95 of *S. album* (Figure 1). Based on the results of the first experiment, the medium with the least nutritional input (i.e., BM0 medium) was used in this experiment. After 30 days, it was observed that the plants had reached approximately the same height and had developed the same number of leaves as in the first experiment (Table 3).

The results of the two experiments showed that the micropropagation strategy was reproducible using axillary buds of a given size (i.e., 0.5 cm, with 1 or 2 axillary buds) and that uniform plant development was achieved in each culture cycle (Figure 2). Based on these results, we estimated the micropropagation potential for these ecotypes over six months, taking into account that each culture cycle had a duration of 30 days and that the multiplication rate of ecotypes EC5 of *S. sediforme* and EC95 of *S. album* was 5, while that of ecotypes EC59 of *S. sediforme* and EC85 of *S. album* was 6 (see Figure 1). Therefore, starting from a plant cultivated in vitro, in ecotypes EC5 and EC95, 5 plants would be obtained in one culture cycle, 25 in two cycles, 125 in three, and following this geometric progression, at the end of the sixth growing cycle (i.e., after 6 months), 15,625 plants would have been obtained. For ecotypes EC59 and EC85, 46,656 plants could be obtained in 6 culture cycles.

We also wanted to calculate the number of plants we could grow on a shelving unit in the plant growth chamber of our laboratory. It should be noted that we could cultivate 5 to 10 Sedum plants in each culture vessel (Appendix A). On the other hand, the shelving unit of the culture chamber in our laboratory consists of 3 shelves with a surface area of 1.92 m^2^ (i.e., 120 cm wide × 160 cm long). Therefore, 221 culture vessels could be incubated on each shelf, which means 1105 plants if 5 plants per vessel were cultivated or 2210 plants in the case of cultivating 10 plants per vessel. In short, each shelving unit can hold 3315 plants (5 axillary shoots/vessel) or 6630 plants (10 axillary shoots/vessel), representing an area of 6.76 m^2^. These results reflect the high propagation potential of this technology.

### 2.3. Multiplication of S. sediforme and S. album by Regeneration Via Somatic Organogenesis from Leaf Explants

#### 2.3.1. Effect of Plant Growth Regulators (PGRs) on Adventitious Shoot Regeneration

In the commercial propagation of plants, the ease with which plant material can be multiplied is a substantial advantage. Many of the plants belonging to the Crassulaceae family can multiply quickly from their leaves. In plants where this occurs, the union between the leaf and the plant is weak, allowing them to detach easily and generate new plants when they fall on a suitable substrate [28].

One of the in vitro culture techniques for the mass production of plants consists of growing leaves on a sterile culture medium containing mineral salts, carbon sources, vitamins, and PGRs. Through a process of somatic organogenesis, this technique leads to the differentiation of buds that, after a process of development and elongation, can originate shoots that, when grown on a rooting medium, give rise to whole plants. In this work, we observed that culturing *S. album* leaves on a medium supplemented with an auxin (i.e., naphthaleneacetic acid) and a cytokinin (i.e., 6-benzyladenine) resulted in bud differentiation (Figure 3a). However, in this culture medium, *S. sediforme* leaves formed only unorganized callus (Figure 3b). In a subsequent trial, the absence of auxin in the culture medium was found to promote bud differentiation in *S. sediforme*.

To verify these results, we cultured leaves of ecotypes EC5 and EC59 of *S. sediforme* and ecotypes EC85 and EC95 of *S. album* on NB10/20 and B20 media, supplemented with auxin and cytokinin or cytokinin alone, respectively. Consistent with what we had observed previously, the leaves of the two ecotypes of *S. sediforme* did not develop buds on the auxin- and cytokinin-supplemented medium but experienced excessive unorganized callus development. In contrast, on media supplemented with cytokinin alone, *S. sediforme* leaf explants developed adventitious buds (Table 4). The two ecotypes of *S. album* developed buds in both culture media, in the medium supplemented with auxin and cytokinin and in the medium supplemented only with cytokinin. However, the best results were obtained in the medium combining auxin and cytokinin (Table 4).

Several scientific studies have shown that the hormonal requirements for callus induction or plant regeneration vary among different species of the genus *Sedum*. For example, the plant growth regulators (PGRs) used for plant regeneration Via somatic organogenesis from leaf explants of *S. sarmentosum* [11] and *S. alfredii* [13] are the same. In both studies, the authors employed a combination of 2,4-dichlorophenoxyacetic acid (2,4-D) and benzyladenine (BA) to induce callus formation, and a combination of naphthaleneacetic acid (NAA) and BA to promote shoot formation. However, in *S. sarmentosum*, the optimal callus induction was achieved with 3.0 mg/L 2,4-D and 1.0 mg/L BA, whereas in *S. alfredii*, the best results were obtained with 1.0 mg/L 2,4-D and 0.5 mg/L BA. Notably, using 3.0 mg/L 2,4-D in *S. alfredii* resulted in a very low callus induction frequency per explant [13]. The callus of *S. aizoon* L. can be induced in MS medium supplemented with 2.0 mg/L 2,4-D and 0.5 mg/L BA [17]. The results indicate that the concentration of 2,4-D plays a more critical role in callus induction than BA. Accordingly, in media containing 2.0 mg/L of 2,4-D, no significant differences are observed between media supplemented solely with 2,4-D and those containing a combination of 2.0 mg/L 2,4-D with either 0.5 or 1.0 mg/L BA. However, significant differences are observed when the 2,4-D concentration drops below 2 mg/L. In *S. spectabile* Boreau, Thidiazuron (TDZ) was found to be more effective for shoot induction than BA [14]. In fact, BA did not yield positive results for shoot regeneration in this species. In *S. dasyphyllum* L., different concentrations of BA or TDZ alone or combined with a-naphthaleneacetic acid (NAA) were tested to stimulate multiple shoot production. In this species, shoot induction and maximized shoot numbers were obtained on explants cultured on media supplemented with 2 μM BA and 1 μM NAA combinations [16]. However, according to the authors, optimal levels of BA (2 μM) combined with higher concentrations of NAA (2 and 4 μM) significantly decreased the number of shoots produced per explant because the explants developed more callus than shoots [16]. Kitamura et al. investigated the response of stem and leaf explants of *S. drymarioides* cultured in media containing varying concentrations of NAA and BA [9]. Their results indicated that leaf explants exhibited a higher organogenic potential than stem explants, with the most efficient organ differentiation observed in media supplemented with 1 mg/L NAA and 10 mg/L BA [9]. These and other studies on *Sedum* species demonstrate that optimal organogenic structure formation is achieved in media with a higher cytokinin-to-auxin ratio [9,16]. In the case of *S. album*, our results are consistent with those reported for *S. dasyphyllum* and *S. drymarioides*, as the best responses were obtained in media supplemented with both NAA and BA. In contrast, this combination of plant growth regulators (PGRs) was ineffective in inducing organogenesis in leaf explants of *S. sediforme*. It has also been reported that BA was more effective than TDZ in promoting shoot production per explant in *S. sarmentosum* Bunge [12]. In *S. praeltum* A. DC., optimal indirect shoot regeneration is achieved with BA concentrations ranging from 0.5 to 1.0 mg/L [15]. Using higher BA concentrations or combinations with 2,4-D significantly reduces shoot regeneration in this species. Similarly, in *S. sediforme*, the culture of leaf explants on medium supplemented with BA resulted in the induction of bud differentiation. However, the simultaneous application of NAA and BA inhibited bud formation and favored the proliferation of disorganized callus tissue.

On the other hand, regarding *S. sediforme*, the best results were obtained with ecotype EC59, in which the percentage of leaves that developed buds was significantly higher than in ecotype EC5 (Table 4). Even so, the percentage of explants that developed buds in ecotype EC5 was greater than 50%. Regarding *S. album*, the rate of explants developing buds was similar in the two ecotypes in the medium supplemented with the two hormones. We verified that ecotype EC85 of *S. album* also developed buds in the medium supplemented only with cytokinin. In this regard, the percentage of explants developing buds in ecotype EC85 was statistically similar in the two culture media. However, in ecotype EC95 of *S. album*, the percentage of explants developing bud was very low in the medium supplemented only with cytokinin, and results of interest were obtained only in the medium supplemented with the two hormones (Table 4). These results are consistent with those related to multiplication from axillary shoot culture, where the highest-yielding ecotypes were EC59 of *S. sediforme* and EC85 of *S. album*.

#### 2.3.2. Proliferation and Elongation of Adventitious Shoots

Induction of adventitious buds from leaves is achieved after 30 days (Figure 4a). The organogenic zone (i.e., callus with buds) is then separated from the rest of the leaf and cultured on a culture medium with the same combination of growth regulators (i.e., NB 1.0/2.0 or B 2.0) for a period of another 30 days (Figure 4b). The objective is to amplify the development of adventitious buds. For the micropropagation strategy based on somatic organogenesis to be efficient, a common practice is the establishment of permanent organogenic lines (callus maintenance). In this way, the number of callus from which several shoots will potentially develop is geometrically amplified. Multiple shoot differentiation using a proliferation–maintenance culture medium has been successfully implemented in barley using meristematic shoot segments as starting explants [29]. Repetitive somatic embryogenesis by reculturing somatic embryos in a maintenance medium has also been successfully used in onion [30].

A preliminary experiment was carried out in which, after obtaining organogenic callus, each of them was divided into two parts: one of which was cultivated in a “shoot elongation medium” (Figure 4c) and the other in a “maintenance medium” (Figure 4d). The maintenance medium used was the most favorable for inducing adventitious bud development in each ecotype (i.e., NB 1.0/2.0 for S album and B 2.0 for *S. sediforme*). Organogenic callus transferred to the maintenance medium increased in size and developed new adventitious buds within 30 days. For shoot elongation, the concentration of each hormone was significantly reduced. Specifically, NB 0.1/0.1 culture medium was used for *S. album* and B 0.1 culture medium for *S. sediforme*.

Many studies have reported that the medium used for shoot induction is the same as that for shoot regeneration [11,14]. In other cases, organogenic structures are transferred to a culture medium without cytokinin [12]. For the elongation of *S. alfredii*, Zhao et al. found that gibberellic acid (GA_3_) was required, with the best results obtained in a medium supplemented with 3 mg/L GA_3_ [13]. In this study, we did not completely eliminate cytokinin but significantly reduced its concentration. This elongation medium facilitated the development of shoots that could be isolated and transferred to a rooting medium.

We determined the number of shoots that could be generated from each callus after 30 days of culture on the elongation medium. As can be seen in Table 4, all four ecotypes were able to generate around four shoots from each organogenic callus. We counted the number of shoots developed on each callus during two new culture cycles and found that a similar number of shoots per callus elongated in each cycle. Based on the results obtained, the micropropagation potential of each of the ecotypes was determined. Regardless of the number of leaves that can be obtained from a plant that has grown for 30 days in the medium for Sedum cloning (i.e., BM0, see Table 2 and Table 3), we opted for prudence when making the calculations and established that only 12 leaves would be used from each plant (Table 5). Using this type of explant as starting material, during the first two months, the induction of adventitious buds (Figure 4a) and the amplification of organogenic structures (Figure 4b) would be carried out. However, shoot elongation occurs from the third month on (Figure 4c), resulting in approximately 27, 38, 35, and 32 plants of ecotypes EC5, EC57, EC85, and EC95, respectively (Table 5).

#### 2.3.3. Rooting of Adventitious Shoots

Shoots that develop from an organogenic callus (Figure 4c) have no root system. Therefore, when individualized from the callus, they must be cultured in a medium that promotes rooting. In our case, based on the previous results (see Table 2 and Table 3), plants regenerated by somatic organogenesis of the four ecotypes were cultured on BM0 cloning medium.

Different rooting behaviors have been observed among *Sedum* species during in vitro rooting. For instance, in *S. dasyphyllum*, shoots exhibited optimal rooting on half-strength MS medium supplemented with 2 μM indole-3-butyric acid (IBA) [16]. Similarly, in *S. alfredii*, shoots rooted more effectively on medium containing IBA, with the highest root formation observed at 2.0 mg/L IBA [13]. However, the percentage of rooted shoots was very low when the medium was supplemented with various concentrations of indole-3-acetic acid (IAA) or naphthaleneacetic acid (NAA) [13]. In *S. sarmentosum*, shoots rooted equally well on hormone-free medium and medium supplemented with IBA, whereas root induction was minimal or absent on NAA-supplemented medium [12]. In *S. spectabile*, root induction and the average number of roots were higher on auxin-free medium than auxin-containing medium [14].

We observed that *S. sediforme* and *S. album* rooted very well on hormone-free medium. Root development occurred in plants of all four ecotypes, although the root system of *S. album* ecotypes was different from that of *S. sediforme* (Figure 5). Thus, *S. sediforme* developed fewer roots of greater length (Figure 5a,b), while *S. album* developed abundant roots but of short length (Figure 5c,d). In both cases, roots were thin and did not cover much surface area. In general, higher root development was observed in ecotype EC59 of *S. sediforme* and EC85 of *S. album*, which is consistent with the better results of these two ecotypes in multiplication experiments.

### 2.4. Acclimation of Axenic Plants of S. sediforme and S. album

One of the most critical phases in a micropropagation process is acclimatization since, in general, plants grown in vitro are very susceptible to environmental changes. Indeed, during the first ex vitro stages, the vitro-plants need protection to avoid rapid dehydration. A simple practice for acclimatization of vitro plants is to protect them with a plastic cup [31,32]. The plastic cup is gradually removed to achieve a progressive adaptation to ex vitro growth conditions.

*Sedum* species have a CAM, meaning they can thrive easily in arid regions. These species minimize water loss by closing their stomata during the day and absorbing CO_2_ at night [33]. This metabolic process may help ease the shift from the stable and regulated environment of in vitro cultivation to the more dynamic and unpredictable conditions of ex vitro growth, promoting a smoother adaptation. Indeed, it is well established that plants cultivated under in vitro conditions often exhibit a range of physiological and morphological abnormalities, among which impaired stomatal function is particularly prominent. Due to this anomaly, in vitro-propagated plants are highly prone to wilting when transferred from the controlled culture environment to ambient atmospheric conditions, making the acclimatization process especially challenging. The relative humidity within culture vessels typically approaches 100%, and prolonged exposure to such high humidity, common in micropropagation, can significantly disrupt stomatal function and increase cuticular permeability [34]. Consequently, upon transfer to ex vitro conditions, these plants initially require elevated humidity levels to ensure survival. However, Santamaría and colleagues observed that the stomata of micropropagated *Agave tequilana* (Weber), a species with CAM, responded to various factors in a manner similar to that of greenhouse-grown plants [35]. According to the authors, in vitro culture does not impair the leaves’ ability to regulate water loss, nor does it alter the characteristic nocturnal stomatal opening typical of this CAM species [35]. Likewise, unlike other plant species, cacti with CAM obtained in vitro can exhibit high resilience during acclimatization. In the cactus species *Obregonia denegrii* Fric. and *Coryphantha minima* Baird, water loss during the transition to ex vitro conditions does not constitute a critical stress factor [36]. In fact, in *kalanchoe beharensis* it has been demonstrated that the acclimatization process can be carried out by exposing the plant directly to ex vitro conditions without the need to protect it with a plastic cup [32].

In this study, we acclimatized 100 plants derived from in vitro culture of *Sedum sediforme* (50 plants from ecotype EC5 and 50 from ecotype EC59) and 100 plants of *Sedum album* (50 from ecotype EC85 and 50 from ecotype EC95). Half of the plants were covered with plastic cups for each ecotype, while the other half remained uncovered to assess the feasibility of acclimatization without plastic cup protection. The plastic cups were removed seven days after transferring the plants from in vitro to ex vitro conditions in the covered group. Upon evaluation, we found no noticeable differences between the uncovered plants and those initially protected with plastic cups (Table 6, Figure 6). After two months, our results indicated that *Sedum* plants cultivated in vitro were generally capable of transitioning to ex vitro conditions without requiring a microclimate of higher relative humidity, demonstrating successful acclimatization without the need for plastic cup protection.

## 3. Materials and Methods

### 3.1. Plant Material and In Vitro Introduction of S. sediforme and S. album Plants

As starting plant material, we used two ecotypes of *S. sediforme* and two ecotypes of *S. album* that were previously collected from different geographical areas of the Valencian Community. These ecotypes were coded with an abbreviation (i.e., EC) and a number. Ecotypes EC5 and EC59 of *S. sediforme* were collected in Ribesalves (coordinates 39°35′00″ North, 0°30′45″ West) and Benitandus (39°55′41″ North, 0°20′11″ West), respectively, in December 2016, while ecotypes EC85 and EC95 of *S. album* were collected in Agres (38°46′33″ North, 0°30′56″ West) and Sagunto (39°40′32″ North, 0°16′39″ West), respectively, in January 2017. After collection, they were propagated by staking and maintained in the Instituto Universitario de Conservación y Mejora de la Agrodiversidad Valenciana (COMAV) greenhouses. The plants in the greenhouse were subjected to natural photoperiods, receiving sunlight under environmental conditions that varied seasonally. Temperature ranged from 18 ± 2 °C during winter to 38 ± 2 °C in summer, while relative humidity was maintained between 30% and 50%.

Top stem segments of approximately 4 cm in length, including several axillary buds, were used to disinfect the plant material (Appendix A). Previously, these stems were carefully washed with tap water and 0.1% (*v*/*v*) Tween 20 surfactant (Sigma-Aldrich, Burlington, MA, USA) (Appendix A). For surface sterilization, the stem segments were treated in a laminar flow cabinet for 20 min with a 20% domestic bleach solution (*v*/*v*; 4% sodium hypochlorite) supplemented with 0.1% of the surfactant Tween-20 (*v*/*v*). Then, the disinfectant solution was removed by three successive washes (5, 10, and 15 min, respectively) with sterile distilled water (Appendix A).

Once disinfected, the nodal explants were obtained. For this purpose, on sterile filter paper and in a laminar flow cabinet, the stem segments were cut into smaller nodal explants containing 1–2 axillary buds after removing a small portion of each end. Subsequently, the nodal explants were cultured in 300 mL culture vessels with 40 mL of basal medium (BM0) (Table 7). The cultures were incubated under controlled environmental conditions in a climatic chamber at 25° C under a 16 h/8 h (day/night) photoperiod with fluorescent light (photosynthetic photon flux of 100 μmol m^−2^ s^−1^).

### 3.2. Multiplication by Axillary Shoot Culture

Multiplication of the four *Sedum* ecotypes was evaluated on BM0, BM1, and BM2 media (Table 7). In each medium, 30 axillary shoots were used as starting material for each ecotype. For this purpose, nodal explants of approximately 1 cm with at least one axillary bud were cut from 30-day-old axenic plants. Nodal explants were grown vertically in 300 mL culture vessels containing 40 mL of culture medium. All cultures were incubated under controlled environmental conditions in a climatic chamber, as previously described. The height of axenic plants and the number of leaves developed by axenic plants after 30 days of cultivation were recorded.

### 3.3. Multiplication by Regeneration Via Somatic Organogenesis from Leaf Explants

(a)Induction of adventitious buds

NB10/20 and B20 media (see Table 8) were used to evaluate somatic organogenesis from leaf explants. These culture media were used because, in a preliminary experiment, we found that this concentration was adequate to induce adventitious regeneration on leaves of *S. sediforme* and *S. album* [37]. In each medium, 40 explants were used as starting material for each ecotype. Leaves were excised from one-month-old axenic plants. Once the leaves were separated from the plant, the two ends were removed and cultured with the abaxial surface in contact with the culture medium. The explants were cultured in 90 mm Petri dishes (10 explants per dish) containing 20 mL of culture medium. All cultures were incubated under controlled environmental conditions in a climatic chamber, as previously described. The frequency of explants developing adventitious buds at 30 days of culture was recorded.

(b)Proliferation of the organogenic zone

After 30 days of culture, organogenic zones were excised from the leaf fragment that had not developed morphogenesis and cultured on fresh culture medium with the same combination of PGRs (i.e., NB1.0/2.0 or B2.0, Table 8). The explants were cultured in 90 mm Petri dishes (10 explants per dish) containing 20 mL of culture medium. All cultures were incubated under controlled environmental conditions in a climatic chamber, as previously described.

(c)Shoot elongation

For shoot elongation, culture media with lower cytokinin concentration were used. Specifically, organogenic callus grown on NB1.0/2.0 medium were transferred to NB0.1/0.1 medium (Table 8), and organogenic callus grown on B2.0 medium were transferred to B0.1 medium (Table 8). Organogenic calli were grown in 300 mL culture vessels containing 40 mL of culture medium. All cultures were incubated under controlled environmental conditions in a climatic chamber, as previously described. The average number of shoots generated from each organogenic callus was recorded.

(d)Rooting of adventitious shoots

Individual shoots were excised and transferred to BM0 (Table 7). Shoots were grown vertically in 300 mL culture vessels containing 40 mL of culture medium. All cultures were incubated under controlled environmental conditions in a climatic chamber, as previously described.

### 3.4. Statistical Analysis

Statistical analyses were conducted using Statgraphics Centurion XVIII software (Statgraphics Technologies Inc., The Plains, VA, USA). Given the non-normal distribution of the data, the Kruskal–Wallis non-parametric test was employed to assess overall group differences. Post hoc pairwise comparisons were subsequently performed using Conover’s method, with statistical significance established at a threshold of *p* < 0.05.

### 3.5. Acclimatization of Axenic Plants

For the acclimatization process, the axenic plants were carefully removed from the culture vessels and placed on absorbent paper to eliminate excess agar from the roots, taking care to avoid root damage. Roots were then gently rinsed with water, and the plants were transplanted into plastic pots containing coconut fiber. For each ecotype, 50 plants were used: half (25 plants per ecotype) were covered with a transparent plastic cup to increase humidity, while the other half remained uncovered. All plants were placed in growth chambers equipped with a Phytotron System, programmed to maintain a constant temperature of 25 °C and a 16/8 h light/dark photoperiod. During the acclimatization period, irrigation with water and Hoagland’s solution were alternated to keep the substrate hydrated.

## 4. Conclusions

The main objective of this study has been to establish the first micropropagation protocol for *S. sediforme* and two of *S. album*, providing a rapid and space-saving alternative to conventional propagation techniques. To achieve this, we implemented a straightforward methodology for the in vitro introduction of the plant material. Through axillary shoot culture, we developed a protocol capable of producing thousands of axenic plants from an axillary bud within six months. In addition, a method based on somatic organogenesis from leaf explants was optimized. The leaf explants of *S. album* formed buds on media supplemented with auxins and cytokinins, whereas *S. sediforme* responded only to cytokinin-enriched media. A significant reduction in cytokinin concentration was necessary to induce elongation of adventitious shoots. These shoots are rooted efficiently on a hormone-free basal medium. The final acclimatization phase was successfully performed ex vitro without protective measures against dehydration, thus lowering the overall cost of the process.

## Figures and Tables

**Figure 1 plants-14-01819-f001:**
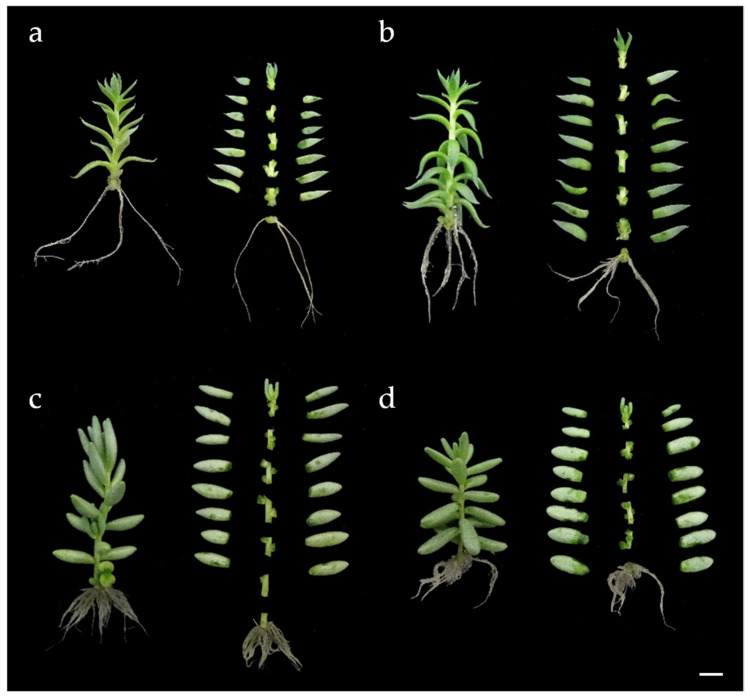
Axenic plants of ecotypes EC5 (**a**) and EC59 (**b**) of *S. sediforme* and ecotypes EC85 (**c**) and EC95 (**d**) of *S. album*. Scale bars: 0.5 cm.

**Figure 2 plants-14-01819-f002:**
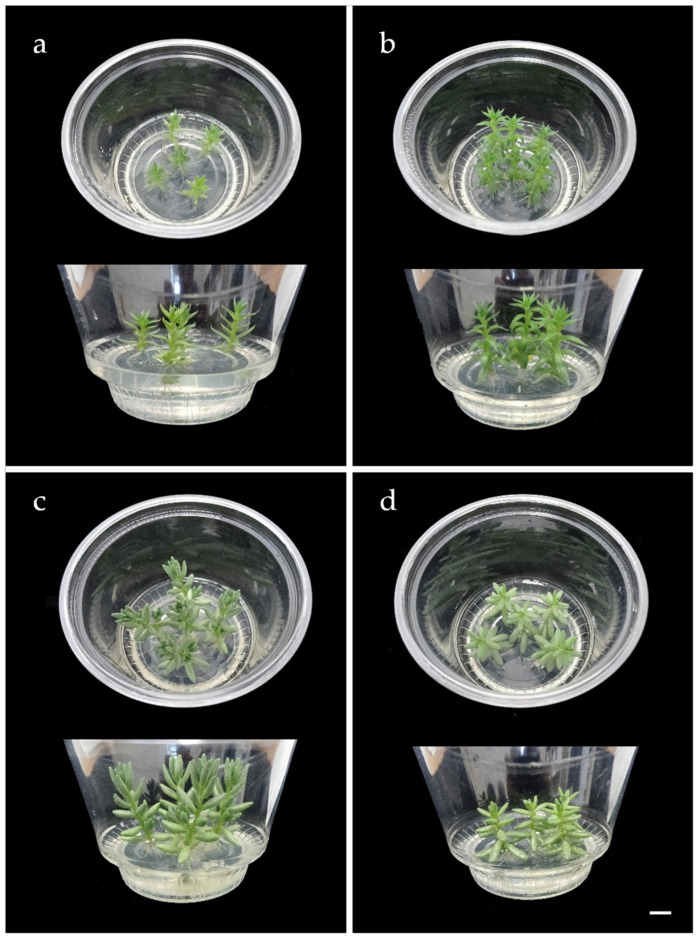
Axenic plants of ecotypes EC5 (**a**) and EC59 (**b**) of *S. sediforme* and ecotypes EC85 (**c**) and EC95 (**d**) of *S. album* after one cycle of in vitro culture (30 days). Scale bars: 1 cm.

**Figure 3 plants-14-01819-f003:**
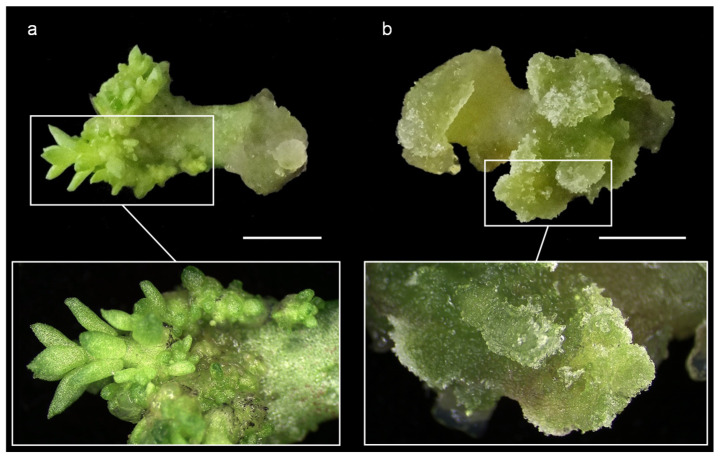
Development of adventitious buds and unorganized callus in leaf explants of *S. album* (**a**) and *S. sediforme* (**b**) grown on medium supplemented with auxin and cytokinin. Scale bars: 0.5 cm.

**Figure 4 plants-14-01819-f004:**
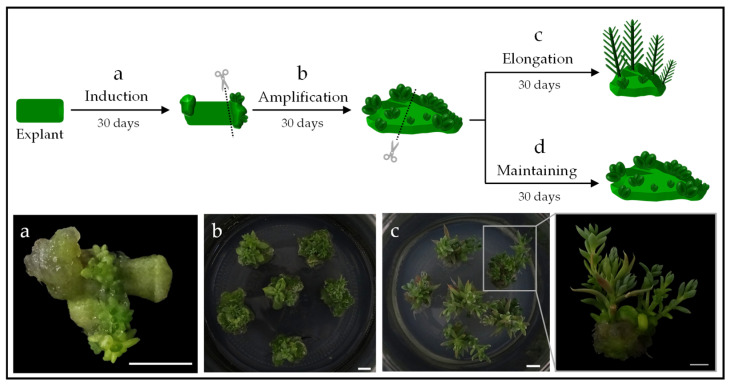
Micropropagation of *S. sediforme* by somatic organogenesis from leaf explants. (**a**) Adventitious bud induction, (**b**) callus development and adventitious bud amplification, (**c**) shoot elongation and close-up of shoot elongation (photo bottom right), and (**d**) maintenance of permanent organogenic lines. Scale bars = 0.5 cm.

**Figure 5 plants-14-01819-f005:**
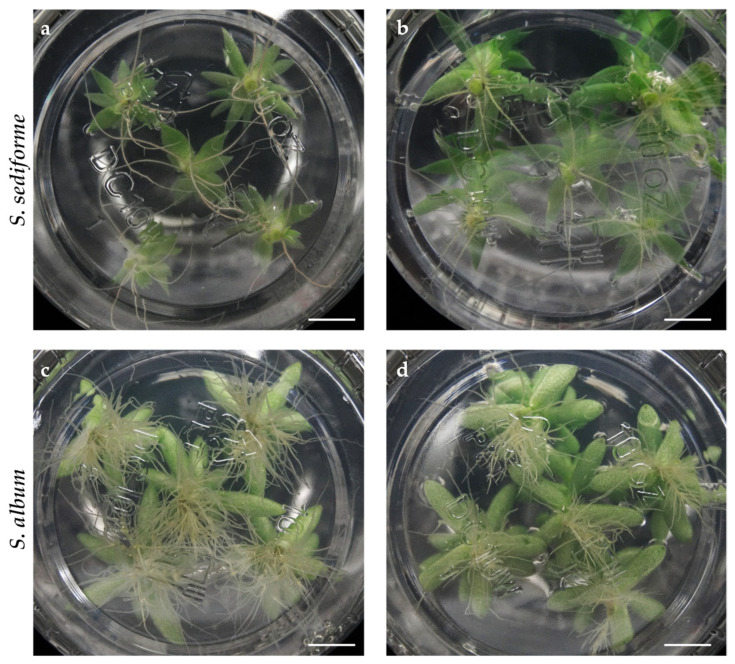
Root system of *S. sediforme* and *S. album* plants after 30 days of culture on MB0 medium. (**a**) Ecotype EC5, (**b**) ecotype EC59, (**c**) ecotype EC85, and (**d**) ecotype EC95. Scale bars: 1 cm.

**Figure 6 plants-14-01819-f006:**
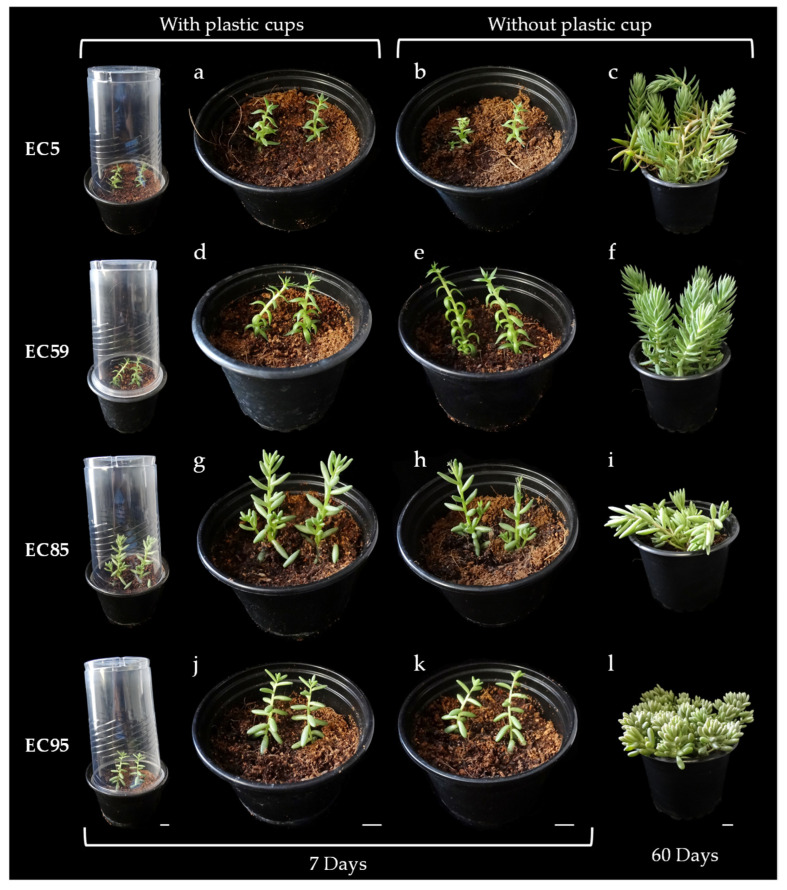
Acclimatization of Sedum axenic plants with and without plastic cups. Developmental status at 7 days (**a**,**b**,**d**,**e**,**g**,**h,j**,**k**) and 60 days (**c,f**,**i**,**l**) after transfer to ex vitro conditions. Comparison of 7-day-old plants acclimatized with (**a,d,g**,**j**) and without (**b**,**e**,**h**,**k**) plastic cups. Scale bars = 1 cm.

**Table 1 plants-14-01819-t001:** Introduction of *S. sediforme* and *S. album* plants under in vitro culture conditions. Average number of successfully sterilized plants in each culture vessel and percentage of sterilized plants.

Species	Ecotype	Successful Introduction by Vessel	Sterilization Success Rate
*S. sediforme*	EC5	4.55 ± 0.31	64.94 ± 4.46
	EC59	2.91 ± 0.25	41.56 ± 3.58
*S. album*	EC85	3.27 ± 0.30	46.75 ± 4.35
	EC95	4.27 ± 0.30	61.04 ± 4.35

The values are given as the mean ± SE. Of each ecotype, 77 shoots were disinfected at a rate of 7 shoots per culture vessel, as described in Section 3.1 of Materials and Methods.

**Table 2 plants-14-01819-t002:** Height and number of leaves of *S. sediforme* and *S. album* plants cultivated for 30 days in vitro on different culture media.

Species	Ecotype	Culture Medium	Plant Height (cm)	No. of Leaves
*S. sediforme*	EC5	MB2	3.37 ± 0.05 (a)	18.20 ± 0.86 (b)
		BM1	3.38 ± 0.04 (a)	18.73 ± 0.77 (b)
		BM0	3.46 ± 0.04 (a)	19.20 ± 0.67 (b)
	EC59	MB2	4.14 ± 0.06 (c)	21.60 ± 0.43 (c)
		BM1	4.19 ± 0.04 (c)	21.86 ± 0.39 (c)
		BM0	4.23 ± 0.03 (c)	22.07 ± 0.34 (c)
*S. album*	EC85	MB2	4.18 ± 0.04 (c)	21.20 ± 0.46 (c)
		BM1	4.21 ± 0.04 (c)	21.40 ± 0.42 (c)
		BM0	4.24 ± 0.03 (c)	21.66 ± 0.32 (c)
	EC95	MB2	3.76 ± 0.05 (b)	16.13 ± 0.41 (a)
		BM1	3.68 ± 0.06 (b)	15.80 ± 0.26 (a)
		BM0	3.81 ± 0.05 (b)	16.33 ± 0.23 (a)

The values are given as the mean ± SE. Means for each parameter followed by different letters are significantly different according to Kruskal–Wallis test (*p* ≤ 0.05).

**Table 3 plants-14-01819-t003:** Height and number of leaves of *S. sediforme* and *S. album* plants cultivated for 30 days in vitro on MB0 culture medium.

Species	Ecotype	Plant Height (cm)	No. of Leaves
*S. sediforme*	EC5	3.55 ± 0.03 (a)	19.33 ± 0.50 (b)
	EC59	4.21 ± 0.03 (b)	21.86 ± 0.36 (c)
*S. album*	EC85	4.29 ± 0.03 (b)	22.20 ± 0.29 (c)
	EC95	3.66 ± 0.06 (a)	15.80 ± 0.26 (a)

The values are given as the mean ± SE. Means for each parameter followed by different letters are significantly different according to Kruskal–Wallis test (*p* ≤ 0.05).

**Table 4 plants-14-01819-t004:** Effect of growth regulators on the frequency of explants developing adventitious buds and the average number of shoots developing in each leaf explant.

Species	Ecotype	PGR (mg/L)		Frequency of Explants with Buds	No. of Shoots/Leaf Explant
		NAA	6-BA		
*S. sediforme*	EC5	0.0	2.0	57.81% ± 4.37 (b)	4.04 ± 0.16 (b)
		1.0	2.0	0.00% ± 0.00 (a)	0.00% ± 0.00 (a)
	EC59	0.0	2.0	79.81% ± 3.56 (d)	4.24 ± 0.16 (b)
		1.0	2.0	0.00% ± 0.00 (a)	0.00% ± 0.00 (a)
*S. album*	EC85	0.0	2.0	64.91% ± 4.47 (bc)	4.12 ± 0.17 (b)
		1.0	2.0	73.81% ± 3.92 (cd)	4.16 ± 0.19 (b)
	EC95	0.0	2.0	4.69% ± 0.95 (a)	n.d.
		1.0	2.0	66.94% ± 4.28 (bc)	4.08 ± 0.16 (b)

The values are given as the mean ± SE. Means for each parameter followed by different letters are significantly different according to Kruskal–Wallis test (*p* ≤ 0.05).

**Table 5 plants-14-01819-t005:** Number of plants that would be obtained in only three months (induction → amplification → elongation) from 12 leaves of one plant.

Species	Ecotype	No. Explants (a)	Frequency of Explants with Buds (b)	No. Callus (c)	No. of Shoots/Leaf Explant (d)	No. Plants(e)
*S. sediforme*	EC5	12	57.81%	6.93	4	27.74
	EC59	12	79.69%	9.56	4	38.25
*S. album*	EC85	12	73.83%	8.86	4	35.42
	EC95	12	66.97%	8.03	4	32.11

Note: the number of plants (e) is determined by the number of leaves to be used from each plant (a), the frequency of explants with buds from that cultivar (b, see Table 4), the number of organogenic callus to be obtained from the explants with buds (c) and the average number of plants regenerating from each callus (d, see Table 4).

**Table 6 plants-14-01819-t006:** Acclimatization response of *S. sediforme* and *S. album* axenic plants under different acclimatization conditions: protected with a transparent plastic cup vs. unprotected.

Species	Ecotype	No. Plants / Acclimatization Condition	Frequency of Acclimatized Plants (%)
*S. sediforme*	EC5	25 / protected with plastic cups	84%
		25 / unprotected without plastic cups	80%
	EC59	25 / protected with plastic cups	92%
		25 / unprotected without plastic cups	88%
*S. album*	EC85	25 / protected with plastic cups	92%
		25 / unprotected without plastic cups	96%
	EC95	25 / protected with plastic cups	92%
		25 / unprotected without plastic cups	92%

**Table 7 plants-14-01819-t007:** Composition of culture media used for rooting and axillary shoot development.

Compounds	BM0	BM1	BM2
MS *^1^ solutions	50%	100%	100%
Myo-inositol * (mg/L)	100	100	100
Sucrose * (g/L)	10	10	20
Thiamine–HCl * (mg/L)	1	1	1
European bacteriological agar ** (g/L)	7	7	7

* Duchefa Biochemie, Haarlem, The Netherlands. ** Condalab, Torrejón de Ardoz, Madrid, Spain (batch number: AO1886, CAT. 1800.05). ^1^ [26].

**Table 8 plants-14-01819-t008:** Composition of culture media used for induction, proliferation, and elongation of adventitious shoots.

Compounds	B2.0	NB1.0/2.0	B0.1	NB0.1/0.1
MS *^1^ solutions	100%	100%	100%	100%
Myo-inositol * (mg/L)	100	100	100	100
Sucrose * (g/L)	30	30	30	30
SH ^2^ vitamins	100%	100%	100%	100%
NAA * (mg/L)	-	1	-	0.1
BA * (mg/L)	2	2	0.1	0.1
European bacteriological agar ** (g/L)	7	7	7	7

* Duchefa Biochemie, Haarlem, The Netherlands. ** Condalab, Torrejón de Ardoz, Madrid, Spain (batch number: AO1886, CAT. 1800.05). ^1^ [26]. ^2^ [38].

## Data Availability

Data are contained within the article.

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
