# Peer review of "Efficient Micropropagation of Sedum sediforme and S. album for Large-Scale Propagation and Integration into Green Roof Systems"

_plants, 2025, doi:10.3390/plants14121819_

Round 1
Reviewer 1 Report
Comments and Suggestions for Authors
Title. In vitro micropropagation is redundant because micropropagation is performed in vitro, therefore it is not necessary to write in vitro micropropagation, just write micropropagation. “Micropropagation of Sedum sediforme and S. album for Large-Scale Propagation and Integration into Green Roof Systems”
Introduction
Justify the necessary in vitro propagation of the two species Sedum sediforme and S. album since they are not in danger of extinction, threatened or their vegetative propagation is slow or unlikely, since in vitro propagation requires the investment of technical and economic resources.
Write "in vitro" in italics throughout the text
There are publications on micropropagation of Sedum species that should be in the background.
Park HY, Saini RK, Gopal J, Keum YS, Kim DH, Lee O, Sivanesan I. Micropropagation and Subsequent Enrichment of Carotenoids, Fatty Acids, and Tocopherol Contents in Sedum dasyphyllum L. Front Chem. 2017 Oct 9;5:77. doi: 10.3389/fchem.2017.00077
Yoshie Kitamura, Kouji Kubo Laiq ur Rahman, Toshihiko Ikenaga. Reproduction of Sedum drymaroides an endangered rare species, by micropropagation. Plant Biotechnology, 19 (5), 303-309 (2002)
Material and Methods
Confirm whether the "basal medium (BM0)" is the MS culture medium (Murashige and Skoog, 1962); if so, include the citation in the text.
Results and Discussion
One of the objectives of in vitro propagation of S. sediforme and S. album is for large-scale propagation and integration into green roof systems, but there is no data on large-scale propagation or integration into green roof systems, so this part of the study is missing.
Compare the results obtained in the in vitro propagation of the two species studied (S. sediforme and S. album) with the data reported for S. dasyphyllum and S. drymaroides
Park HY, Saini RK, Gopal J, Keum YS, Kim DH, Lee O, Sivanesan I. Micropropagation and Subsequent Enrichment of Carotenoids, Fatty Acids, and Tocopherol Contents in Sedum dasyphyllum L. Front Chem. 2017 Oct 9;5:77. doi: 10.3389/fchem.2017.00077
Yoshie Kitamura, Kouji Kubo Laiq ur Rahman, Toshihiko Ikenaga. Reproduction of Sedum drymaroides an endangered rare species, by micropropagation. Plant Biotechnology, 19 (5), 303-309 (2002)
References. Update citations
doi: 10.3389/fchem.2017.00077
Plant Biotechnology, 19 (5), 303-309 (2002)
Author Response
The corrections are highlighted in grey text
Comments 1: Title. In vitro micropropagation is redundant because micropropagation is performed in vitro, therefore it is not necessary to write in vitro micropropagation, just write micropropagation. “Micropropagation of Sedum sediforme and S. album for Large-Scale Propagation and Integration into Green Roof Systems”
Response 1: We agree with the reviewer. As suggested, we have removed “in vitro” from the title.
Comments 2: Justify the necessary in vitro propagation of the two species Sedum sediforme and S. album since they are not in danger of extinction, threatened or their vegetative propagation is slow or unlikely, since in vitro propagation requires the investment of technical and economic resources.
Response 2: We thank the reviewer for this valuable suggestion. The authors concur that it is important to highlight the significance of developing a micropropagation protocol for these two Sedum species, particularly given the absence of prior studies on this topic in the scientific literature. In response to Reviewer 2’s recommendation, we have revised the introduction to emphasize the relevance of micropropagation while streamlining the sections related to green roofs. Within this revised context, we have provided a rationale for the importance of establishing micropropagation protocols based on regeneration pathways, specifically axillary and somatic organogenesis.
Comments 3: Write "in vitro" in italics throughout the text
Response 3: We appreciate the reviewer's advice. We have written “in vitro” in italics throughout the text.
Comments 4: There are publications on micropropagation of Sedum species that should be in the background.
Response 4: We appreciate the reviewer's suggestion. In the introduction, we have included a comment on the two studies highlighted by the reviewer, concerning the micropropagation processes of other Sedum species, and in the results and discussion section, we have compared our findings with the results reported in those studies.
Material and Methods
Comments 5: Confirm whether the "basal medium (BM0)" is the MS culture medium (Murashige and Skoog, 1962); if so, include the citation in the text.
Response 5: As can be seen in Table 7, BM0 medium is not exactly MS culture medium. This culture medium is composed of half-strength MS solution, myo-inositol, sucrose, and thiamine–HCl. The citation (Murashige and Skoog, 1962) can be found in the footnotes to the table.
Results and Discussion
Comments 6: One of the objectives of in vitro propagation of S. sediforme and S. album is for large-scale propagation and integration into green roof systems, but there is no data on large-scale propagation or integration into green roof systems, so this part of the study is missing.
Response 6: We sincerely regret any confusion the title of our manuscript may have caused. The primary objective of this study was to develop a micropropagation protocol for two Sedum species that are among the most commonly used in extensive green roof systems. To address the potential for large-scale propagation, we estimated the number of plants that could be produced over six culture cycles (equivalent to six months) starting from a single in vitro plant. Based on our calculations, it would be possible to obtain 15,625 plants from ecotypes EC5 and EC95, and 46,656 plants from ecotypes EC59 and EC85. While we did not complete six full culture cycles, we conducted four cycles and were able to confirm the reproducibility of the protocol, achieving yields of 625 and 1,296 plants for EC5/EC95 and EC59/EC85, respectively. Given the geometric nature of this progression, the projected six-cycle output appears realistic. Although we have not yet tested these micropropagated plants in green roof applications, we compared them with conventionally propagated plants following the acclimatization process and found no phenotypic differences. If the reviewer believes a change in the title is warranted to more accurately reflect the content and avoid any misunderstanding, we would be happy to revise it accordingly.
Comments 7: Compare the results obtained in the in vitro propagation of the two species studied (S. sediforme and S. album) with the data reported for S. dasyphyllum and S. drymaroides
Response 7: Done. As suggested by the reviewer, we have compared our results with those reported for S. dasyphyllum and S. drymaroides.
Reviewer 2 Report
Comments and Suggestions for Authors
This study develops micropropagation protocols of two native ecotypes of S. sediforme and S. album from the Valencian Community. A micropropagation system by axillary buds and a protocol for adventitious organogenesis from leaves were developed. The work in general is well presented, although in some points it needs improvement, material and methods are adequate, but the introduction and discussion need revision (please see comments below).
Line (L) 31: please do not include words showing in the title as kewwords ( i.e., Sedum sediforme, Sedum album)
L 41, 44: Please note that there are a number of more recent reviews that have a broader theme than referring to "tropical climates" like the one you are referring to, i.e., [1]. Kamarulzaman, N.; Hashim, S.Z.; Hashim, H.; Saleh, A.A. Green Roof Concepts as a Passive Cooling Approach in Tropical Climate- An Overview. E3S Web of Conf. 2014, 3, 01028. http://dx.doi.org/10.1051/e3sconf/20140301028
L 70: the phrases “the Crassulacean acid metabolism (CAM) photosynthetic pathway, their root system is superficial” needs adjustment to be grammatically correct according to the following phrases
The introduction needs to be extensively revised. Reduce the length of the introduction on green roofs and add information (references) on in vitro propagation of Sedum spp. or other succulents. Your work is on micropropagation of sedums while your introduction is on green roofs! There are many works published on sedums micropropagation/tissue culture. And you should point out what new is providing the present work.
L 94, 109 please use italics for Latin names in 2.1. section (and in Line 302)
I recommend to use “surface sterilization” instead of disinfection all over manuscript
L 120 -123 :you state: “ Similarly, it has been documented that due to bacterial and fungal contamination during nodal segment culture, the in vitro establishment of switchgrass (Panicum virgatum L.) requires the addition of PPM as a biocide and Benomyl as a fungi-cide in the incubation solution [18].” The reference to this work is unnecessary; the plant species used in it has nothing to do with the smooth, easily sterilized leaves of SedumHundreds of references can be found regarding implant surface sterilization, if you are going to refer to one, find a relevant one.
In section 2.2. and (Table 2) please report how many shoots were developed per explant. From figure 2 I understand you were taking one shoot per explant. The explants used were apical or nodal? From figure 2 I understand were both. In line 199 you say “These results reflect the high propagation potential of this technology”, but this may not be true, because efficient micropropagation protocols often give much higher proliferation per explant, and culture vessels (as yours) with the number of explants you used per vessel occupy the same space in the growth room as in your case.
L204-207: please provide references
L 370-380, please discuss the fact that leaf stomata are not functional in vitro and therefore when microplants are transferred ex vitro initially require higher relative humidity to survive. Do succulents have functional stomata immediately after ex vitro transfer? Are there any references on this?
L 152-153….It is well established that rooting of microplants is more successful in ½ MS rather than full MS, so there is no meaning to report “Spren gel, optimal rooting of plantlets has been achieved in a half-strength MS medium [23].” Particularly as a “discussion” on shoot multiplication and not on rooting!
L 414-415: In materials and methods (line 414-415), please provide information on temperature, moisture and irradiation in the green house where the stock plants were kept.
L 446: …”NB10/20 and B20 media (see Table 8)” there is a problem with the way the media are shown on the top of Table 8.

Author Response
The corrections are highlighted in red font
Comments 1: Line (L) 31: please do not include words showing in the title as kewwords ( i.e., Sedum sediforme, Sedum album)
Response 1: We appreciate the reviewer's suggestion. We have removed the two words that appeared in the title and added the word “crassulaceae”.
Comments 2: L 41, 44: Please note that there are a number of more recent reviews that have a broader theme than referring to "tropical climates" like the one you are referring to, i.e., [1]. Kamarulzaman, N.; Hashim, S.Z.; Hashim, H.; Saleh, A.A. Green Roof Concepts as a Passive Cooling Approach in Tropical Climate- An Overview. E3S Web of Conf. 2014, 3, 01028. http://dx.doi.org/10.1051/e3sconf/20140301028
Response 2: We thank the reviewer for this valuable suggestion. In response to one of the reviewer’s suggestions, we have substantially revised the introduction to place greater emphasis on the central topic of the article: the micropropagation. We believe these changes have significantly enhanced the clarity and focus of the introductory section. We have cited the reference http://dx.doi.org/10.1051/e3sconf/20140301028, as it offers a clear and concise distinction between intensive and extensive green roofs. Building on this framework, we included additional commentary on extensive green roofs, which allowed us to establish a logical connection to the species under study and, subsequently, to the micropropagation process.
Comments 3: L 70: the phrases “the Crassulacean acid metabolism (CAM) photosynthetic pathway, their root system is superficial” needs adjustment to be grammatically correct according to the following phrases
Response 3: We appreciate the reviewer's advice. We have rewritten that paragraph to make it grammatically correct.
Comments 4: The introduction needs to be extensively revised. Reduce the length of the introduction on green roofs and add information (references) on in vitro propagation of Sedum spp. or other succulents. Your work is on micropropagation of sedums while your introduction is on green roofs! There are many works published on sedums micropropagation/tissue culture. And you should point out what new is providing the present work.
Response 4: We thank the reviewer for this valuable suggestion. As previously mentioned, and in accordance with the reviewer’s recommendation, we have substantially reduced the length of the section devoted to green roofs. In its place, we have added information on micropropagation, providing a more comprehensive overview of the technique in general and more specific information related to the micropropagation of Sedum species. Additionally, we have made an effort to highlight the novelty of our study by incorporating a dedicated paragraph that underscores its original contributions to the existing body of knowledge. We hope these adjustments have improved the focus and scientific relevance of the manuscript.
Comments 5: L 94, 109 please use italics for Latin names in 2.1. section (and in Line 302)
Response 5: Done (L106-118)
Comments 6: I recommend to use “surface sterilization” instead of disinfection all over manuscript
Response 6: We appreciate the reviewer's advice. We have replaced “disinfection” with “surface sterilization” throughout the manuscript.
Comments 7: L 120 -123 :you state: “ Similarly, it has been documented that due to bacterial and fungal contamination during nodal segment culture, the in vitro establishment of switchgrass (Panicum virgatum L.) requires the addition of PPM as a biocide and Benomyl as a fungi-cide in the incubation solution [18].” The reference to this work is unnecessary; the plant species used in it has nothing to do with the smooth, easily sterilized leaves of SedumHundreds of references can be found regarding implant surface sterilization, if you are going to refer to one, find a relevant one.
Response 7: We fully agree with the reviewer's statement. We have removed this sentence and the associated reference.
Comments 8: In section 2.2. and (Table 2) please report how many shoots were developed per explant. From figure 2 I understand you were taking one shoot per explant. The explants used were apical or nodal? From figure 2 I understand were both. In line 199 you say “These results reflect the high propagation potential of this technology”, but this may not be true, because efficient micropropagation protocols often give much higher proliferation per explant, and culture vessels (as yours) with the number of explants you used per vessel occupy the same space in the growth room as in your case.
Response 8: We are sincerely grateful to the reviewer for giving us the opportunity to clarify the content of Table 2 and Figure 2. Table 2 presents a comparative analysis of plant height and leaf number in axenic plants derived from axillary shoots cultured over one month in three different media (BM0, BM1, and BM2; for composition details, see the Materials and Methods section). The main objective of this table is to demonstrate that the growth parameters, plant height and leaf number, did not vary significantly across the different media. These findings were confirmed through additional experiments, which yielded consistent results. Consequently, we selected the simplest medium, BM0, for further propagation due to its comparable efficacy and ease of use. In axillary shoot-based micropropagation, each axillary shoot gives rise to a complete plant. After one month of cultivation in BM0, each plant typically produces 5–6 new axillary shoots (including one terminal shoot, as shown in Figure 1). When these new shoots are subsequently cultured again in BM0, they in turn develop into 5–6 new axenic plants, each of which has 5-6 axillary buds. This method, axillary shoot cloning, represents one of the most straightforward micropropagation systems, enabling the rapid generation of acclimatizable plants in a short time frame. On the other hand, Figure 2 illustrates the morphology of axenic plants from each ecotype after 30 days of growth in BM0 medium. This technique, based on a geometric progression of axillary shoot multiplication, can yield between 15,625 and 46,656 plants from a single initial plant, depending on the ecotype. We acknowledge the reviewer’s observation that micropropagation systems based on adventitious morphogenesis are generally more prolific. Indeed, such systems are described in detail in Section 2.3 of the manuscript.
Comments 9: L204-207: please provide references
Response 9: Done (L215)
Comments 10: L 370-380, please discuss the fact that leaf stomata are not functional in vitro and therefore when microplants are transferred ex vitro initially require higher relative humidity to survive. Do succulents have functional stomata immediately after ex vitro transfer? Are there any references on this?
Response 10: We appreciate the reviewer's advice. We have incorporated a paragraph addressing the non-functionality of stomata in leaves of plants cultured in vitro. Notably, we identified a study involving a plant with CAM metabolism in which the stomata were shown to remain functional under in vitro conditions. We believe that this discussion, prompted by the reviewer’s suggestion, has significantly enriched and strengthened that section of the manuscript.
Comments 11: L 152-153….It is well established that rooting of microplants is more successful in ½ MS rather than full MS, so there is no meaning to report “Spren gel, optimal rooting of plantlets has been achieved in a half-strength MS medium [23].” Particularly as a “discussion” on shoot multiplication and not on rooting!
Response 11: We agree with the reviewer, so we have removed this reference.
Comments 12: L 414-415: In materials and methods (line 414-415), please provide information on temperature, moisture and irradiation in the greenhouse where the stock plants were kept.
Response 12: Thanks for the advice. We have added the information requested by the reviewer (lines 452-455).
Comments 13: L 446: …”NB10/20 and B20 media (see Table 8)” there is a problem with the way the media are shown on the top of Table 8.
Response 13: We are a bit puzzled by this comment, as we do not observe any issues with the presentation of the media at the top of Table 8 in our version of the file.
Round 2
Reviewer 1 Report
Comments and Suggestions for Authors
Title. In vitro micropropagation is redundant because micropropagation is performed in vitro, therefore it is not necessary to write in vitro micropropagation, just write micropropagation. “Micropropagation of Sedum sediforme and S. album for Large-Scale Propagation and Integration into Green Roof Systems”
Answer. The title of the manuscript has been corrected according to the reviewer's comment.
Introduction
Justify the necessary in vitro propagation of the two species Sedum sediforme and S. album since they are not in danger of extinction, threatened or their vegetative propagation is slow or unlikely, since in vitro propagation requires the investment of technical and economic resources.
Write "in vitro" in italics throughout the text
There are publications on micropropagation of Sedum species that should be in the background.
Park HY, Saini RK, Gopal J, Keum YS, Kim DH, Lee O, Sivanesan I. Micropropagation and Subsequent Enrichment of Carotenoids, Fatty Acids, and Tocopherol Contents in Sedum dasyphyllum L. Front Chem. 2017 Oct 9;5:77. doi: 10.3389/fchem.2017.00077
Yoshie Kitamura, Kouji Kubo Laiq ur Rahman, Toshihiko Ikenaga. Reproduction of Sedum drymaroides an endangered rare species, by micropropagation. Plant Biotechnology, 19 (5), 303-309 (2002)
Answer. The justification for plant micropropagation has been added and the background has been updated.
Material and Methods
Confirm whether the "basal medium (BM0)" is the MS culture medium (Murashige and Skoog, 1962); if so, include the citation in the text.
Answer. The composition of the "basal medium" (BM0) was indicated, it is the MS culture medium
Results and Discussion
One of the objectives of in vitro propagation of S. sediforme and S. album is for large-scale propagation and integration into green roof systems, but there is no data on large-scale propagation or integration into green roof systems, so this part of the study is missing.
Compare the results obtained in the in vitro propagation of the two species studied (S. sediforme and S. album) with the data reported for S. dasyphyllum and S. drymaroides
Park HY, Saini RK, Gopal J, Keum YS, Kim DH, Lee O, Sivanesan I. Micropropagation and Subsequent Enrichment of Carotenoids, Fatty Acids, and Tocopherol Contents in Sedum dasyphyllum L. Front Chem. 2017 Oct 9;5:77. doi: 10.3389/fchem.2017.00077
Yoshie Kitamura, Kouji Kubo Laiq ur Rahman, Toshihiko Ikenaga. Reproduction of Sedum drymaroides an endangered rare species, by micropropagation. Plant Biotechnology, 19 (5), 303-309 (2002)
Answer. Large-scale propagation or integration into green roof systems has been clarified and the results of micropropagation have been compared with other similar studies.
References
Update citations
doi: 10.3389/fchem.2017.00077
Plant Biotechnology, 19 (5), 303-309 (2002)
Answer. The citations have been updated
Reviewer 2 Report
Comments and Suggestions for Authors
Dear authors,
thank you for your response to my comments in a very satisfactory way. I recommend that your article be published as is.
Best regards